# Quality Control after Intracochlear Intralabyrinthine Schwannoma Resection and Cochlear Implantation

**DOI:** 10.3390/brainsci11091221

**Published:** 2021-09-16

**Authors:** Holger Sudhoff, Lars Uwe Scholtz, Hans Björn Gehl, Ingo Todt

**Affiliations:** 1Department of Otolaryngology, Head and Neck Surgery, Medical School OWL, Bielefeld University, Klinikum Bielefeld Mitte, Teutoburger Str. 50, 33604 Bielefeld, Germany; Holger.Sudhoff@rub.de (H.S.); lars-uwe.scholtz@klinikumbielefeld.de (L.U.S.); 2Department of Radiology, Medical School OWL, Bielefeld University, Klinikum Bielefeld Mitte, 33604 Bielefeld, Germany; hans-bjoern.gehl@klinikumbielefeld.de

**Keywords:** MRI, cochlear implant, intracochlear intralabyrinthine schwannoma

## Abstract

Background: The combination of intralabyrinthine schwannoma (ILS) removal and cochlear implantation is the standard of care after surgical resection for audiological rehabilitation. Intracochlear ILS is not only the most frequent tumor in this group of schwannomas, but it is also, to some degree, surgically the most challenging because of its position behind the modiolus. Recent developments in the knowledge of implant position, implant magnet choice, and magnetic resonance imaging (MRI) sequences make an MRI follow-up after surgical removal possible. Thus far, no results are known about the surgical success and residual rate of these kind of tumors. The aim of the present study was to perform an early MRI follow-up for the evaluation of residual or recurrent intracochlear ILS after surgical removal and cochlear implantation. Methods: In a retrospective study, we evaluated seven patients after an intracochlear ILS removal and single-stage cochlear implantation with a mean period of 13.4 months post surgery with a 3T T1 GAD 2 mm sequence for a residual ILS. Patients were operated on using an individualized technique concept. Results: In six out of seven cases, 3 T T1 GAD 2 mm MRI follow-up showed no residual or recurrent tumor. In one case, a T1 signal indicated a tumor of the upper inner auditory canal (IAC) at the MRI follow up. Conclusion: MRI follow-up as a quality control tool after ILS removal and cochlear implantation is highly important to exclude residual tumors. Long-term MRI evaluation results are needed and can be obtained under consideration of implant position, implant magnet, and MRI sequence choice. A preoperative MRI slice thickness less than 2 mm can be recommended to visualize possible modiolar and IAC expansion.

## 1. Introduction

Cochlear implant is the treatment of choice in cases of sensorineural hearing loss (SNHL) and deafness which cannot be treated by hearing aids. Even in cases of SNHL in addition or related to a vestibular schwannoma, cochlear implantation has been shown to be a possible, and in many cases successful, treatment option [1,2]. First performed in 1999 [3] for intralabyrinthine schwannomas (ILSs), additional cochlear implantation is the treatment of choice for hearing rehabilitation. In such cases, tumor removal and cochlear implant are usually performed as a single-stage surgery. Hearing rehabilitation results are, in most cases, comparable to a regular cochlear implantation outcome but are limited by the extent and position of the ILS [4,5,6,7]. Kennedy et al. classified these tumors in terms of their location. The authors differ between cochlear, vestibulocochlear, vestibular, transmodiolar, and further different subtypes of ILS [8,9]. Based on tumor position, different techniques have been described for performing individualized tumor localization-dependent surgical approaches. Within this group of tumors, intracochlear tumors are special because they are visually covered by the modiolus and are therefore more complicated to remove than other types of tumors. Specifically, for these tumors, different removal techniques have been described [5,10]. Endoscopic techniques have been proposed as minimally invasive approaches [11].

Long-term control of the surgical success of the tumor removal and a follow-up have so far not been possible. The new developments in cochlear implant magnet development, implant position, and head position inside the scanner make postoperative magnetic resonance imaging (MRI) follow-up of this tumor class possible [12]. Second generation magnet (e.g., diametral bipolar), positioning of the receiver implant magnet more than 8 cm from the external auditory canal, low artifact generating sequences (e.g., T2 TSE, non 3D), and a chin-to-chest position of the head inside the scanner are the key factors.

This follow-up acts as a quality control of the tumor removal in the short-term perspective and can provide long-term information about the rate of recurrence/residual growth rate of these tumors. The initial classification as a slow-growing tumor based on a very small number of cases is controversial and did not consider cases of electrical stimulation (cochlear implant) [13,14,15,16,17].

The aim of the present observation was to evaluate specifically an early MRI follow-up after intracochlear ILS removal and cochlear implantation to control the surgical success of tumor surgery.

## 2. Material and Methods

In this study, 7 patients underwent an MRI scan in a tertiary referral center after surgical removal of an intracochlear ILS or combined vestibulocochlear ILS and were included. Patients with a pure vestibular ILS were not included in this study. Initial surgery was performed between October 2017 and December 2019. MRI was performed on average 13.4 months after surgery. Four patients were implanted with a MED-EL SYNCHRONY implant (MED-EL, Innsbruck, Austria) with a diametrically magnetized internal magnet. In the other 3 cases, a Cochlear 512 Profile, a Cochlear 622 Profile Plus (Cochlear, Sydney, Australia ), and an Advanced Bionics (AB) 3D System (Advanced Bionics, Stäfa, Swiss), respectively, were implanted. In all cases, a single-stage surgery was performed.

In all cases, the implant magnet was intraoperatively determined and positioned 8–9 cm behind the external auditory canal [18].

MRI examinations were performed without a headband in the MED-EL SYNCHRONY, Cochlear 622, and AB 3D cases using the 3 T MRI unit (Achieva, Philips Medical Systems, Philips, Amsterdam, The Netherlands). A headband was used in the Cochlear 512 case. For this case a 1.5 T MRI unit (Achieva, Philips Medical Systems) was used.

### 2.1. Surgical Approach

Surgical access was individually chosen, applied to the tumor extent, and microscopically performed. In cases 3 and 4, a test electrode was used to push/pull the electrode out of the second turn [5]. In this technique, a test electrode is used to mobilize the tumor from its post-modiolar attachment and is pushed/pulled out of the cochlea. In cases 5 and 6, a Gelfoam pusher technique was applied.

In one case, a drill-out was performed [19,20]. The drill-out technique is performed as a two-cavity technique with a decapping of the bone from the scalar turns. After placement of the electrode, it is covered with facia, cartilage, and bone dust.

In 3 cases, access to the cochlea was combined with a labyrinthectomy related to a vestibular extension of the ILS. In 2 cases, a wide cochleostomy was used to access the tumor. In 4 cases, wide cochleostomy access was combined with access to the second turn [21]. In one of these cases, a tumor growth under the stapes footplate was removed.

In 2 cases, a modification of the technique was used. Along with a regular cochleostomy and an additional second turn approach, partial removal of the tumor through the basal and second turn was performed. To remove the post-modiolar part of the tumor, a tight roll of dry Gelfoam was introduced into the basal turn (Figure 1a). This was pushed into the scala until the residual ILS was visible and removable, about the second turn (Figure 1b). Then, the Gelfoam was flooded with saline (Figure 1c), and after 10 to 15 min, the swollen wet foam could be sucked out (Figure 1d).

### 2.2. Sequence

The 3 T:T1/T1 Gad, slice thickness 2 mm, resolution: 0.6 × 0.8 mm, FOV 150 × 150, TE 70, TR 3000, TSE 8, Sham Spir.

The 1.5 T:T1/T1 Gad, slice thickness 3 mm, resolution: 0.7 × 0.8 mm, FOV 150 × 150, TE 10, TR 650, TSE 10, Sham Spir.

## 3. Results

In all cases, the cochlea and the IAC were visually accessible. In all cases, complete removal of the intracochlear ILS vestibular parts was confirmed by the MRI. The individual pre- and postoperative MRI T1 Gad figures are exemplary for case 4 and 6 (Figure 2 and Figure 3). In Figure 4b, an initially radiologically missed or residual tumor at the IAC fundus from the vestibulum (transmacular) is shown (case 5), as well as an additional signal in the basal turn.

In all cases, hearing rehabilitation was successful (Table 1). In one case (no. 7), rehabilitation could not be evaluated related to the loss of the patient. 

## 4. Discussion

This series is the first to perform a systematic MRI-based tumor follow-up to evaluate the initial surgical success of an individualized approach to the post-modiolar location of intracochlear ILS. Our findings show that ILS can persist or occur after surgical removal of ILS and underline the need for an MRI follow-up after a surgical removal.

Cochlear implantation is a tool for hearing rehabilitation for patients with vestibular schwannoma [1,22] and ILS [3]. Although audiological results of hearing rehabilitation in cases of vestibular schwannoma cochlear implantation vary [2], cochlear implantation in cases of ILS is mostly successful [4,5,6,10]. Different techniques have been described to enable an individualized approach to the different locations of the tumor, ranging from a pure labyrinthectomy to a labyrinthectomy in addition to a double approach to the cochlea, removal around an enlarged cochleostomy, access around an enlarged cochleostomy combined with second turn access, and drill-out [4,6,10].

Whereas cases of a singular tumor in the vestibulum or an affection of the semicircular canals are easy to surgically remove, cases with involvement of the area behind the modiolus are not visually accessible.

Therefore, we performed three techniques to remove these intracochlear or combined vestibulocochlear ILSs in our study following an individualized approach: removal around an enlarged cochleostomy, combined with second turn access, and a drill-out [10]. For the cases of a combined approach, dummy electrodes [4] and hooks are used. Visual control by micro-endoscopes has been described [11] but was not performed in our study.

In two cases, we performed a Gelfoam pusher technique. This technique consisted of pushing dry Gelfoam rolled inside the scala, pushing the residual tumor to the second turn opening. Afterward, the Gelfoam was expanded with saline and suctioned out (Figure 4a–d).

Hearing outcomes are similar to the results of other study groups (Table 1) [4,5,6,10].

Thus far, there are no data evaluating the success of these different techniques or residual rate of these tumors. ILSs are classified as slow-growing entities [13,14,15]. However, this statement, based on a limited number of patients, can be controversially discussed [16,17] and does not consider electrical stimulation by a cochlear implant.

MRI follow-up after vestibular schwannoma removal is the international standard clinical pathway in vestibular-schwannoma-treating clinics. Walton et al., [23] were the first to perform a schwannoma follow-up after cochlear implantation in NF II cases. The ability to reproducibly evaluate vestibular schwannomas and ILSs after removal and cochlear implantation has recently been shown [12]. Evaluation of the IAC and cochlea after cochlear implantation is based on proper implant and patient position [18,23,24,25]; magnet development, which allows for pain-free observation [26]; and the choice of the right MRI sequence [27]. These points allow for an artifact-minimized visualization of IAC and cochlea. The explantation of the magnet is theoretically possible but clinically, related to the risk of infection, rarely performed.

Even other sequences based on a MARS (material artifact reduction sequences) protocol might alternatively be used to allow a proper visualization of the IAC and the cochlea [27].

Following the assumption of a slow-growing tumor, this series is the first to evaluate the initial surgical success in terms of residual tumors as a quality control.

In this series, through MRI evaluation after a period of 13.4 months, we observed no residual tumor in the area of interest behind the modiolus. In one case, we observed a T1 signal at the upper part of the IAC, indicating a transmacular form. This case needs to be further evaluated in detail to control further growth in the future. For this patient, it remains unclear if the tumor was located in this area before the surgery and was not observed due to the slice thickness of 2 mm or if it was a residual tumor with a fast-growing tendency. The fact that a visual tumor was not observed intraoperatively led us to assume that a partial, very small ingrowth under the radiological visual resolution of 2 mm was persistent preoperatively. Very small areas of ingrowth into the modiolus and the IAC have been described histologically [28]. The additional small signal in the basal turn can be interpreted as a residual tumor.

Based on this finding, a slice thickness thinner than 2 mm must be strongly recommended to limit or exclude missed preoperative information about affections or partial ingrowth of the tumor in the IAC or the modiolus. This important preoperative information would lead to modified surgical planning in this hypothetical case. Especially for clinics with only a 1.5 T scanner, intensive collaboration between the radiology and surgical departments is needed to discuss and optimize sequences to reach a sufficient MRI resolution.

In this single case, we decided against a revision because the audiological result was excellent, and a revision would include the risk of deterioration. In case of a tumor growth, we would reevaluate this decision.

Based on our findings and the known histopathologic literature [28], three locations seem to be of special interest during a follow-up: (1) the post-modiolar region related to an incomplete removal, (2) new transmodiolar ingrowth, and (3) new transmacular ingrowth.

The main limitation of this study is the small number of patients in this cohort. A larger cohort would allow us to further balance the finding. The used resolution of the MRI sequence was not adapted to the findings of the study with a recommendation of a resolution of under 2 mm. This means that a higher resolution might result in a higher number of residual tumors. A long-term MRI follow-up evaluation of this group of patients is necessary and should be integrated in the clinical pathway. Additionally, a long-term evaluation would offer deeper insight into the growth characteristics in cases of residual tumors.

## 5. Conclusions

MRI follow-up as a quality control tool after ILS removal and cochlear implantation is highly important to exclude recurrent or residual tumors. MRI evaluation results are needed as a standard and can be obtained considering implant position, implant magnet, and MRI sequence choice. A preoperative MRI slice thickness less than the used 2 mm should be strongly recommended to visualize modiolar and macular ingrowth; 2 mm might miss important information about the expansion of these small tumors.

## Figures and Tables

**Figure 1 brainsci-11-01221-f001:**
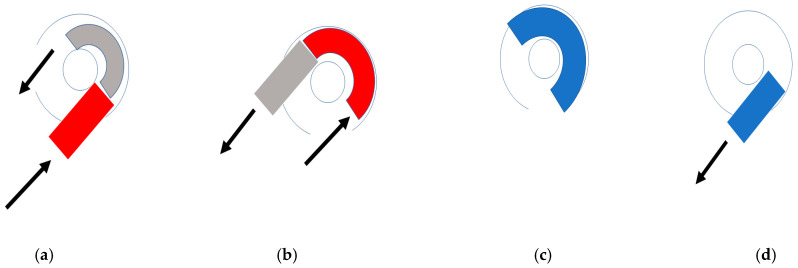
(**a**,**b**) The tumor (grey) behind the modiolus is pushed with a dry roll of Gelfoam (red) from the basal turn to the second turn access (**a**,**b**), until it can be removed at the second turn access. (**c**,**d**) The Gelfoam is rinsed by saline (change from red to blue). It swells and can be sucked out about the basal turn.

**Figure 2 brainsci-11-01221-f002:**
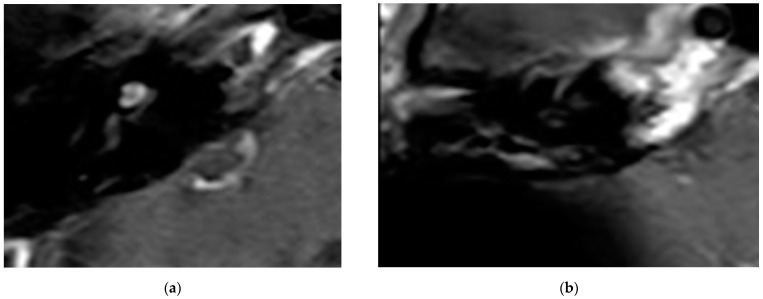
(**a**). An exemplary ILS tumor pre-op (T1, Gad). (**b**). The ILS region with a T1 Gad sequence after removal and CI.

**Figure 3 brainsci-11-01221-f003:**
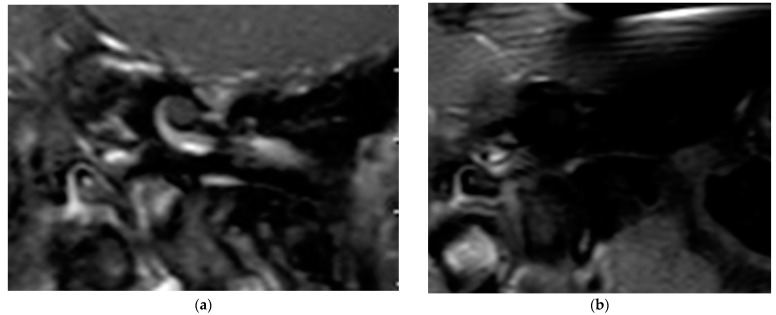
(**a**). The exemplary ILS pre-op (T1, Gad); (**b**). The ILS region with a T1 Gad sequence after removal and CI.

**Figure 4 brainsci-11-01221-f004:**
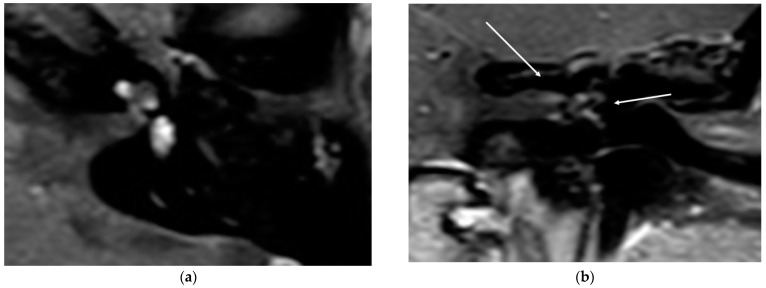
(**a**). An exemplary ILS tumor pre-op (T1, Gad). (**b**). The ILS region with a T1 Gad sequence after removal and CI and detection of a possible tumor in the IAC and in the basal turn.

**Table 1 brainsci-11-01221-t001:** Description of individual data. Period between surgery and control MRI; surgical access to the cochlea (single: extended cochleostomy; double: extended cochleostomy and second turn access; lab.: labyrinthectomy); specific information is about additional techniques performed for tumor removal and anatomic variations; hearing outcome by monosyllabic words quietly at 65 dB.

No.	Month between Surgery and MRI	ILS Form	Surgical Access	Implant	Specific Information	Monosyllabic Understanding at 65 dB
1	3	vestibulo cochlear	cochleostomy	512		70
2	21	cochlear	drill out	Synchrony		30
3	14	cochlear	double access	Synchrony	testelectrode	45
4	19	vestibulo cochlear	labyrinthectomy and double access	622	testelectrode	60
5	15	vestibulo cochlear	labyrinthectomy and double access	Synchrony	gelfoam pusher	55
6	14	cochlear	Double access	AB 3D	gelfoam pusher	65
7	8	cochlear	cochleostomy	Synchrony	apical ossification	Lost follow up

## Data Availability

The original data are available by the corresponding author upon request.

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
