# Peer review of "Quality Control after Intracochlear Intralabyrinthine Schwannoma Resection and Cochlear Implantation"

_brainsci, 2021, doi:10.3390/brainsci11091221_

Round 1

Reviewer 1 Report

The authors describe MRI follow-up in patients after simultaneous removal of an ILS and CI. Imaging was possible in all patients. This finding is relevant to the field.

The MRI was performed on average 13.4 months after surgery. What was the range of follow-up? What was the follow-up time of the patient with the recurrent/residual tumor?

Did you do serial follow-up images or do you intend to do so? For how long?

Line 141: The aim of the study is to evaluate MRI follow up after simultaneous ILS removal and CI. With only one follow-up MRI, I am not sure if this study is designed to assess surgical success (if defined as non-recurrence of the ILS).

Line 143: This is very much in accordance to a recent histopathological analysis (Bagattini et al. Audiol Neurotol, 2021).

Line 178-179: This is a repetition of line 141ff (see comment above).

Line 184: No new paragraph needed. Additionally, please rephrase the sentence as is has the same beginning as the previous sentence.

Line 201: Reference 28 is missing in the reference list.

Discussion:

The CI may affect the MR imaging quality. Can you comment on the MRI quality as compared to an MRI in patients without CI?

Conclusion:

The recommendation of using a thickness of less than 2mm makes sense. However, this conclusion can not be drawn from your study (you did not investigate different slice thicknesses).

Author Response

Dear reviewer,

please find a point to point answer to the queries.

The authors describe MRI follow-up in patients after simultaneous removal of an ILS and CI. Imaging was possible in all patients. This finding is relevant to the field.

 The MRI was performed on average 13.4 months after surgery. What was the range of follow-up? What was the follow-up time of the patient with the recurrent/residual tumor?

We intend to perform an every year follow up. Individual follow up MRI from surgery is shown in Tab. 1. The suspicous case follow up time period was 15 month (case 5)

Did you do serial follow-up images or do you intend to do so? For how long?

We intend to perform a follow up of 5 years

 Line 141: The aim of the study is to evaluate MRI follow up after simultaneous ILS removal and CI. With only one follow-up MRI, I am not sure if this study is designed to assess surgical success (if defined as non-recurrence of the ILS).

We changed the sentence to:.. initial surgical success..

 Line 143: This is very much in accordance to a recent histopathological analysis (Bagattini et al. Audiol Neurotol, 2021).

Right. See reference 28.

 Line 178-179: This is a repetition of line 141ff (see comment above).

 We changed the sentence to:.. initial surgical success..

Line 184: No new paragraph needed. Additionally, please rephrase the sentence as is has the same beginning as the previous sentence.

Changed and rephrased as: For this patient, it…

 Line 201: Reference 28 is missing in the reference list.

  1. Bagattini M, Quesnel AM, Röösli C. Histopathologic Evaluation of Intralabyrinthine Schwannoma. Audiol Neurootol. 2021;26(4):265-272. doi: 10.1159/000511634. Epub 2020 Dec 22. PMID: 33352553; PMCID: PMC8217404.

Discussion:

The CI may affect the MR imaging quality. Can you comment on the MRI quality as compared to an MRI in patients without CI?

We added this information: This points allow for an artifact minimized visualization of IAC and cochlea. The explantation of the magnet is theoretically possible but clinically, related to the risk of infection, rarely performed.

Conclusion:

The recommendation of using a thickness of less than 2mm makes sense. However, this conclusion can not be drawn from your study (you did not investigate different slice thicknesses).

The reviewer is right. We changed the sentence to: A preoperative MRI slice thickness less than  the used 2 mm can be strongly recommended to visualize modiolar and macular ingrowth; 2 mm might miss important information about the expansion of these small tumors.

Thank you for the helpful comments to improve the manuscript.

Reviewer 2 Report

It is an interesting article on MRI evaluation after schwannoma resection and CI. The study population is very small (7 patients), the study is no more than a small case series report, but it adds little knowledge on the field.

The aim of the study should be better explained in the introduction. The English language is clear .   Some points may be further discussed: Did the authors study the post-resection FLAIR sequences? If they did, what about the intralabyrinthine FLAIR residual hyperintensity that may show after schwannoma resection?    What about the CI body positioning? Did the authors choose a particular position of the CI body to permit a better visualization of the homolateral ponto-cerebellar angle?  Even if MRI is the gold standard in schwannoma follow up, also contrast enhanced CT may be useful? Did the authors think that a flat panel CT (given its absolute spatial definition) may be useful in monitoring those cases? In the surgical approach section the authors should indicate if the approaches were microscopic, endoscopic or combinated.  MRI sequences that have been chosen could be further discussed.    Specific comment will follow: Line 64: double space Lines 80-82: even if the correct citation is reported, the push-pull mechanism with probe electrode could be better specified in the text. As well as Gellfoam pusher technique.  Line 81: the same could be indicated for the drill-out technique.

Author Response

Dear reviewer,

please find a point to point answer to the queries.

It is an interesting article on MRI evaluation after schwannoma resection and CI. The study population is very small (7 patients), the study is no more than a small case series report, but it adds little knowledge on the field.

Thank you.

The aim of the study should be better explained in the introduction.

We changed the sentence to: The aim of the present observation was to evaluate an early MRI follow up after specifically intracochlear ILS removal and cochlear implantation to control the surgical success of tumor surgery.

The English language is clear .   Some points may be further discussed: Did the authors study the post-resection FLAIR sequences? If they did, what about the intralabyrinthine FLAIR residual hyperintensity that may show after schwannoma resection?   

FLAIR sequences were not used.

 What about the CI body positioning? Did the authors choose a particular position of the CI body to permit a better visualization of the homolateral ponto-cerebellar angle? 

In all cases the CI receiver magnet was placed 8-9  cm from the external auditory canal. L73

 Even if MRI is the gold standard in schwannoma follow up, also contrast enhanced CT may be useful? Did the authors think that a flat panel CT (given its absolute spatial definition) may be useful in monitoring those cases?

We got no experience with this technique.

In the surgical approach section the authors should indicate, if the approaches were microscopic, endoscopic or combinated.

All approaches were microscopic. We added the information.

  MRI sequences that have been chosen could be further discussed. 

We added the information: Even other sequences based on a MARS (material artifact reduction sequences) protocol might alternatively be used to allow a proper visualization of the IAC and the cochlea [27].

   Specific comment will follow:

Line 64: double space

Changed

Lines 80-82: even if the correct citation is reported, the push-pull mechanism with probe electrode could be better specified in the text. As well as Gellfoam pusher technique. 

We added: In this technique a test electrode is used to mobilize the tumor from its post modiolar attachment and pushed / pulled out of the cochlea appoaches.

The Gelfoam technique is described in L93-99.

Line 81: the same could be indicated for the drill-out technique.

We added: The drill out technique is performed as a two cavity technique with a decapping of the bone from the scalar turns. After placement of the electrode it is covered with facia, cartilage and bone dust.

Thank you for improving the quality of the manuscript.
